# An Experimental Investigation of the Influence of Flow and Pipe Diameter on the Fire Extinguishing Efficiency of Nitrogen Injection in a Narrow Confined Underground Space

**Guowei Zhang [1,2,\*], Dong Guo [1], Bin Li [1], Zhiwei Zhang [1] and Diping Yuan [2]**

[1] School of Safety Engineering, China University of Mining and Technology, Xuzhou 221116, China
[2] Shenzhen Research Institute, China University of Mining and Technology, Shenzhen 518000, China
\* Correspondence: zgw119xz@cumt.edu.cn

**Abstract:** In this study, an underground pipe gallery was taken as the research subject to explore the influence of nitrogen injection flow rate and pipe diameter on the fire extinguishing efficiency in an underground narrow confined space. A liquid nitrogen fire extinguishing test system for the underground narrow confined space was built. The fire extinguishing time, flame height, temperature, and oxygen concentration under different conditions were recorded by liquid nitrogen fire extinguishing tests, and the variations in the characteristics of these data were analyzed. Furthermore, the fire suppression factor, cooling factor, and asphyxiation factor were introduced to quantify the influence of the nitrogen flow rate and pipe diameter on extinguishing efficiency. According to the results, the fire was effectively extinguished by liquid nitrogen in the underground confined space through asphyxiation as the main fire extinguishing mechanism, and the extinguishing time was about 95.5% less than that in the self-extinguishing test. Although the fire suppression efficiency is positively related to the nitrogen injection flow, the asphyxiation efficiency can be reduced when the flow rate is excessive or too weak. Additionally, the asphyxiation factor and fire suppression factor are highly sensitive to the injection pipe diameter. Therefore, a valuable reference is provided in this study for promoting the future engineering application of liquid nitrogen fire extinguishment.

**Keywords:** underground space; confined space; liquid nitrogen; fire extinguishing systems; extinguishing efficiency

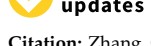



## 1. Introduction

Underground pipe galleries, cable trenches, and other underground facilities are mostly narrow and located in long underground confined spaces that lack sufficient ventilation openings. These spaces have unique fire risks, such as the elevated temperature of the fire site, the problems associated with discharging toxic and harmful gases, the easy occurrence of flashover and reburning, and the difficulties that face fire and rescue personnel trying to enter the fire site [1]. In such spaces, conventional water, dry powder, and other traditional fire extinguishing methods often fail to efficiently extinguish fire [2–5]. Since identifying the fire source within the underground pipe gallery is a challenge and the traditional dense pipelines may block the spraying of a fire extinguishing agent, the agent is unable to act directly on the fire source. In this situation, the fire extinguishing efficiency is reduced. In addition, it takes a long time to remove the residual fire extinguishing agent from a disaster site, which may disrupt the normal functioning of the affected area.

Inert gas is an effective way to address the issues associated with extinguishing a fire in a confined space [6]. In particular, liquid nitrogen is a reliable low-temperature fire extinguishing agent with an expansion ratio of 1:717 at 25 °C. It can generate a large amount of nitrogen and suffocate the fire source within a short time after being injected into the fire area [7], which is especially desirable in underground confined spaces. Once the fire is extinguished, nitrogen, as a non-toxic and harmless gas, can be directly discharged

into the atmosphere through ventilation, which is very conducive to the rapid recovery of facility operations after a disaster. Liquid nitrogen was first used in coal mines in the middle of the last century. In 1953, a coal seam fire near the shaft bottom parking lot in the Roslin Mine in England was extinguished through nitrogen injection, which was one of the earliest applications of liquid nitrogen in a mine fire [8]. As a high-efficiency fire extinguishing agent, liquid nitrogen has also been shown to be outstanding in extinguishing open-space fires, such as metal fires and lithium battery fires [9–11]. Given the similarity between underground and mining spaces, the use of liquid nitrogen appears to be a robust firefighting solution for enclosed underground environments.

The feasibility of using liquid nitrogen to extinguish fires in an underground pipe gallery was verified by the author's research group in 2019 [12]. Later, Li et al. tested liquid nitrogen in an underground pipe gallery and found that vertical downward nitrogen injection had higher fire extinguishing efficiency than horizontal nitrogen injection [13]. Jia et al. used liquid nitrogen in the cable cabin of a comprehensive pipe gallery. It was established that the asphyxiation mechanism of liquid nitrogen was dominant in extinguishing the fire in the cable cabin of the pipe gallery, and the critical oxygen volume fraction was about 15%–16% [14]. Lan et al. reported that ventilation in the confined space enhances the fire extinguishing efficiency of liquid nitrogen, but also makes the smoke spread further within the same time frame [15].

Therefore, liquid nitrogen has great potential for firefighting in enclosed underground environments. However, quantitative research on the fire extinguishing efficiency of liquid nitrogen in underground narrow and long confined spaces is still scarce. In particular, identifying the key factors that affect the fire extinguishing efficiency of liquid nitrogen and determining the impact mechanism of each factor in order to promote the widespread engineering application of liquid nitrogen are among the tasks to be solved. To achieve these goals, an underground pipe gallery was used in the present work as an example of an underground confined space. Special attention was paid to the impact of the nitrogen injection flow and pipe diameter on the fire extinguishing efficiency with regard to fire suppression, cooling, and asphyxiation. The results of this research provide new prospects for fire prevention and the control of underground confined spaces such as underground pipe galleries and cable trenches.

## 2. Performance Indexes for Evaluating Liquid Nitrogen Fire Extinguishing Efficiency

Although the extinguishing efficiency of liquid nitrogen has been widely investigated in recent research, most studies have focused on a rough analysis of the extinguishing needs of different scenarios [16] or experimentally verifying the extinguishing efficiency of liquid nitrogen [17]. Meanwhile, quantitative analyses of the fire extinguishing efficiency of liquid nitrogen under specific fire extinguishing scenarios are still scarce, which makes it difficult to support practical engineering applications. Therefore, an evaluation index of liquid nitrogen fire extinguishing efficiency was developed in this study to clarify the impact of key nitrogen injection parameters on the fire extinguishing efficiency of nitrogen in underground confined spaces.

The performance of a fire extinguishing agent can be directly characterized by the changes in the flame height, temperature, and oxygen volume fraction around the fire during injection of the agent; thus, the corresponding parameters can be used as the criteria to characterize the fire extinguishing efficiency of liquid nitrogen. The fire suppression factor, cooling factor and asphyxiation factor were defined as $a$, $b$, and $c$, which are expressed by the following formulas. They represent the change rate in a certain time in the flame height, temperature directly above the flame, and the oxygen volume fraction around the fire source during nitrogen injection.

$$a = \frac{Flame_{(t_0)} - Flame_{(t)}}{t - t_0} \tag{1}$$

$$b = \frac{Temperature_{(t_0)} - Temperature_{(t)}}{t - t_0} \tag{2}$$

$$c = \frac{Oxygen_{(t_0)} - Oxygen_{(t)}}{t - t_0} \tag{3}$$

where $t_0$ is the starting time of the nitrogen injection and $t$ is the time of the flame extinction. $Flame_{(t_0)}$, $Temperature_{(t_0)}$ and $Oxygen_{(t_0)}$ indicates the flame height, temperature and the oxygen concentration at time $t_0$, respectively. $Flame_{(t)}$ indicates the flame height at time $t$, and so do the other variables. The fire suppression factor $a$, the cooling factor $b$ and the asphyxiation factor $c$ are determined by the flame height, the temperature value and oxygen volume fraction at the beginning and end of the nitrogen injection, respectively.

## 3. Test Design

### 3.1. Test Devices

In order to accurately reproduce a real fire in an underground confined space and explore the characteristics of liquid nitrogen fire extinguishment therein, a liquid nitrogen fire extinguishing test system was built, as shown in Figure 1. The setup consists of three basic components: a reduced-size underground pipe gallery model, a nitrogen injection system, and a data acquisition system.

The underground pipe gallery model is 10.0 m long, and its cross-section is 0.9 m wide and 1.25 m high. The whole model is made of a high-temperature resistant stainless-steel material. Specifically, the top, bottom, and the back are 3 mm thick stainless-steel plates. The front and the rotating doors on both sides of the model are made of fire-resistant glass that can withstand temperatures of hundreds of degrees Celsius, which facilitates the real-time observation of combustion, smoke spread, nitrogen injection and fire extinguishing process during the test.

The main purpose of the test was to obtain the changes in the flame height, temperature, and oxygen volume fraction in an urban underground pipe gallery under the action of liquid nitrogen. Therefore, because of the flame-retardant characteristics and the complex combustion characteristics of cables, a pool fire with an obvious combustion development stage, stable stage and decline stage was selected to simulate the fire scenario in an urban underground pipe gallery. The fire source was a square fuel-pan with a side length of 0.35 m, which was placed in the middle of the pipe gallery model. The fuel selected was n-heptane. The combustion of n-heptane is stable and only produces a small amount of smoke, which is convenient for test observation and flame height recording.

The nitrogen injection system was composed of a liquid nitrogen tank, a Coriolis Mass Flowmeter and a liquid nitrogen transmission pipeline. The liquid nitrogen tank is a self-pressurized heat-insulation tank, and the top is equipped with a liquid outlet valve and a gas outlet valve. The pressure in the tank is displayed on the pressure gauge, and the outlet pressure can be quickly adjusted with the booster valve. The mass flow rate is precisely displayed by the Coriolis Mass Flowmeter in real time. The liquid nitrogen transmission pipeline is a low-temperature and high-pressure resistant stainless-steel hose, wrapped with an insulating cover to reduce the liquid nitrogen consumption and cooling capacity loss in the transmission process. The nitrogen injection was positioned on the ceiling of the pipe gallery, and was about 0.3 m away from the fire source, which not only ensured the effect of extinguishing the fire, but also effectively prevented fuel spillage due to the direct impact of the nitrogen injection on the fire source. The nitrogen injection pipe was inserted vertically downward into the pipe gallery for about 0.1 m.

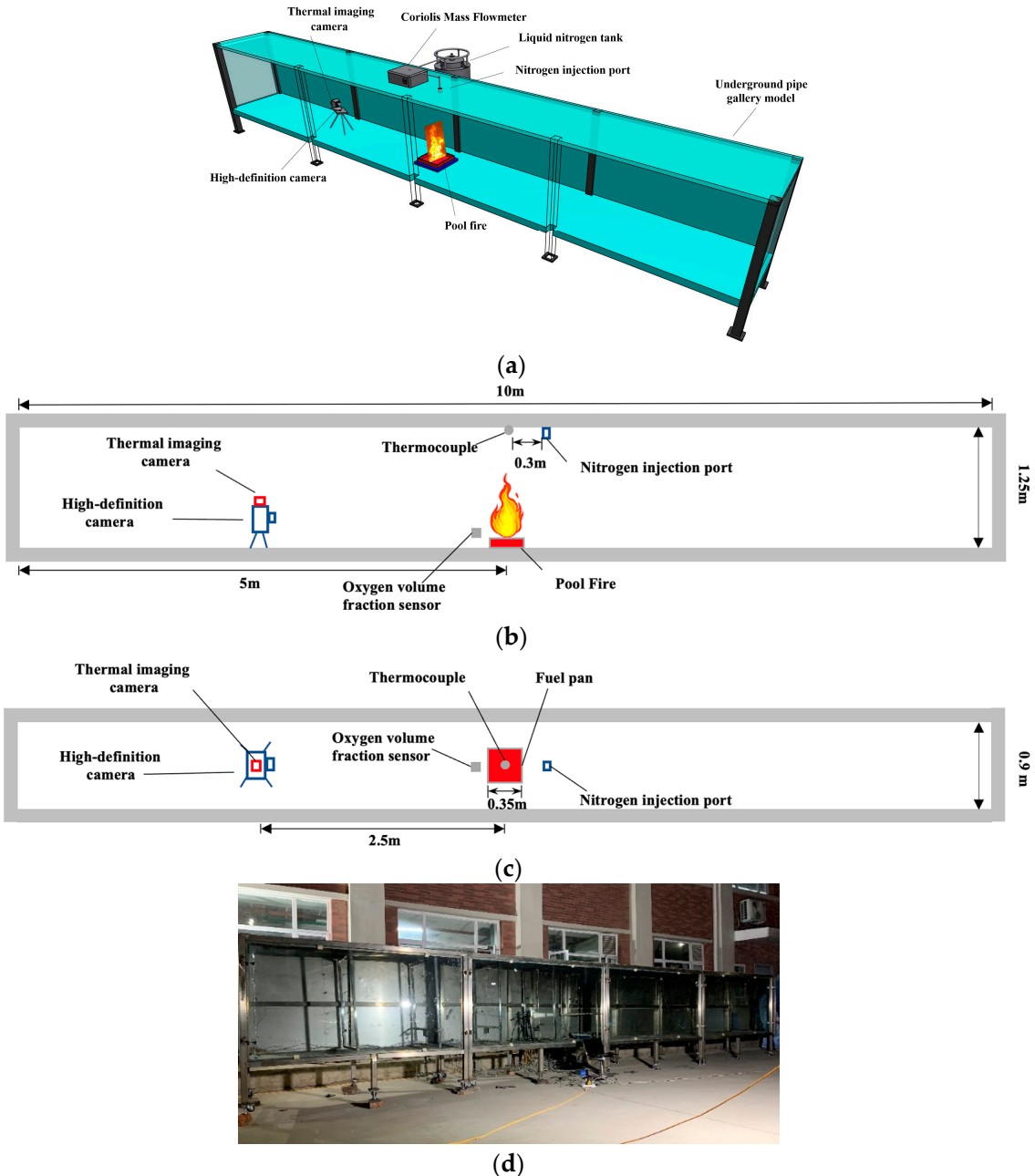

**Figure 1.** The liquid nitrogen fire extinguishing test system for the underground confined space. (**a**) Overall view; (**b**) front view; (**c**) overhead view; (**d**) photo of the underground pipe gallery model.

The data acquisition system included a K-type thermocouple, an oxygen volume fraction sensor, a thermal imaging camera and a high-definition camera. The K-type thermocouple was arranged on the ceiling of the pipe gallery to obtain the temperature change directly above the fire source. The oxygen volume fraction sensor was positioned near the fire source to obtain the change in the oxygen volume fraction, so as to analyze the asphyxiation effect of nitrogen injection on the fire source. In order to prevent high-temperature damage to the sensor, the oxygen volume fraction sensor was placed at the same height as the fire source and was about 0.1 m away from the fire source. The infrared thermal imager and high-definition camera were placed inside the pipe gallery and 2.5 m away from the fire source, to obtain information about the changes in the flame during the test. All data acquisition devices were connected to data lines, which were connected to computers through a hole in the underground pipe gallery model. After the data acquisition

system was tested, all holes and chinks were filled with high-temperature resistant foam material to ensure the airtightness of the underground pipe gallery model.

### 3.2. Test Method

In order to explore the influence of the nitrogen injection flow rate and pipe diameter on the fire extinguishing efficiency of liquid nitrogen, the nitrogen injection flow rate and pipe diameter were selected as the variables to perform the nitrogen injection fire extinguishing test. In the test, the nitrogen injection flow rate was controlled by properly adjusting the degree of opening of the liquid nitrogen tank valve, and the maximum flow rate was 315 kg/h. The pressure of the liquid nitrogen tank was basically maintained at 0.09–0.1 MPa. The diameter of the nozzle used for the nitrogen injection was 10–25 mm. At the same time, the self-extinguishing test results without nitrogen injection were selected as the control group. Three repeated tests were conducted for test condition 2 and the time of the fire extinguishment was 24 s, 23 s and 29 s. Good repeatability of the test result was obtained, and the test with the time of fire extinguishment of 24 s was used for subsequent data analysis. For other tests, one test was conducted. All the test conditions are listed in Table 1.

**Table 1.** Test conditions.

| Test Number | Mode | Distance/m | Diameter/mm | Flow Rate/Maximum Flow Percentage | Remarks |
|---|---|---|---|---|---|
| 1 | - | - | - | - | Control group |
| 2 | | | | 1 | |
| 3 | Vertical Down | 0.3 m away from the fire source | 10 | 3/4 | Change flow rate |
| 4 | | | | 1/2 | |
| 5 | | | | 1/4 | |
| 6 | | | 25 | 1 | |
| 7 | Vertical Down | 0.3 m away from the fire source | 20 | 1 | Change diameter |
| 8 | | | 15 | 1 | |

In the test, 0 s was the time that the test started, and 120 s was the time when the fire entered a fully developed stage. The time of 120 s was selected as the time that the nitrogen injection started ($t_0$). When the flame went out, the time of the flame extinction ($t$) was recorded, the nitrogen injection was stopped immediately, and the test was over. The flame height, temperature, and oxygen volume fraction were recorded during the test.

### 3.3. Test Phenomenon

Taking condition 2 as an example, the flame morphology changes during the test are displayed in Figure 2. At 0 s, the fuel pool was ignited with an extended flame spray gun, and the fire source began to burn violently. As the burning proceeded, the fire gradually expanded and the height of the flame continued to increase. At 10 s, the flame height reached the middle of the pipe gallery (about 0.5 m). At 54 s, the height of the flame exceeded 1 m, which started to affect the ceiling, and the impact frequency gradually increased. Meanwhile, half of the pipe gallery was filled with the smoke. At 120 s, the top of the flame was almost at the same height as the pipe gallery model, and the ceiling was continuously impacted by the flame. The flame and the fuel pool were basically surrounded by the hot smoke, indicating that the fire had entered the fully developed stage. At this time, the nitrogen injection system was turned on to extinguish the fire.

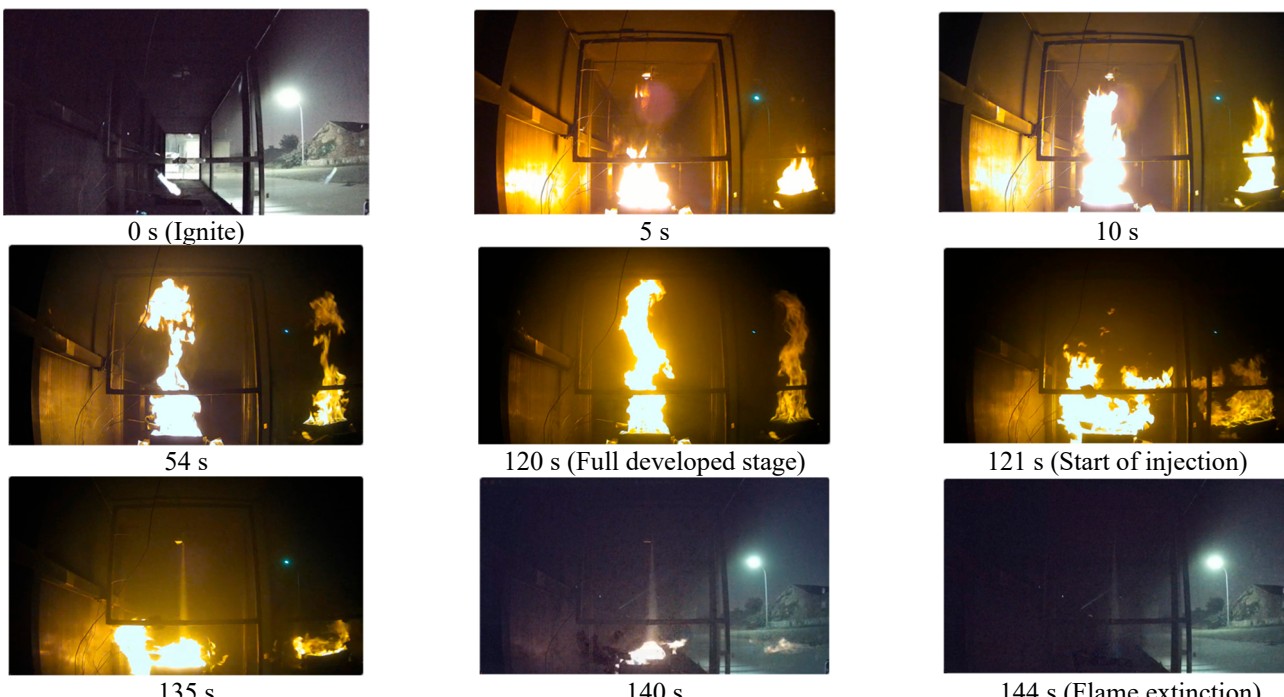

**Figure 2.** Changes in the characteristics and form of the fire under test condition 2.

After the nitrogen injection valve was opened, part of the remaining air in the pipeline was driven at high speed by the nitrogen toward the pipe gallery, strongly disturbing the air flow around the fire source. At 121 s, the flame was inhibited under the impact of nitrogen, and the flame height decreased rapidly, while the flame width and range became larger. At 135 s, the proportion of liquid nitrogen in the injected two-phase nitrogen gradually became greater than that of gaseous nitrogen, and the jet column could be clearly seen at the nitrogen injection port. Under the continuous pressure of the liquid nitrogen on the flame, the flame height was always less than 0.4 m and it fluctuated from side to side below the 0.4 m horizontal line; this lasted for about 6 s. At 144 s, the flame was completely extinguished under the action of the nitrogen injection. At this time, the nitrogen injection valve was closed and the test ended.

## 4. Analysis of the Influence of Nitrogen Injection Flow on Liquid Nitrogen Extinguishing Efficiency

### 4.1. A Comparative Analysis of the Time of Fire Extinguishment

The time of the fire extinguishment was measured from 120 s (the time that the nitrogen injection was started in test conditions 1 to 5) to the time that the flames went out. The time of the fire extinguishment in test conditions 1 to 5 are shown in Table 2. It can be seen that the time of the fire extinguishment under nitrogen injection conditions was significantly reduced compared with that of the control group. For example, while the self-extinguishing time of the flame in the control group was about 246 s, the time of fire extinguishment in test condition 2 was close to 24 s, i.e., it was decreased by 90.2%. Meanwhile, with the decrease in the nitrogen injection flow, the extinguishing time gradually increased. In a certain range of nitrogen injection flow rates, the overall extinguishing efficiency of liquid nitrogen was less affected by the flow rate. However, once the flow rate was reduced to a quarter of the maximum flow rate, the time of the fire extinguishment significantly increased.

**Table 2.** Test results under different test conditions.

| Test Number | Flow Rate /Maximum Flow Percentage | Time of Fire Extinguishment/s | Time$_{1/2}$/s | Remark |
|---|---|---|---|---|
| 1 | - | 246 | 204 | Control group |
| 2 | 1 | 24 | 24 | |
| 3 | 3/4 | 25 | 28 | Change flow rate |
| 4 | 1/2 | 27 | 28 | |
| 5 | 1/4 | 35 | 26 | |

Note: Time$_{1/2}$ is the time when the temperature drops to the half of the maximum temperature.

Furthermore, we found that under different nitrogen injection flow rates, the time for the ceiling temperature to drop to half of the maximum temperature was basically the same, that is, about 24−28 s. By comparing this time with the time of the fire extinguishment, it was found that the two times were very close, which meant that the temperature fell to about the half of the maximum temperature as soon as the fire went out.

### 4.2. A Comparative Analysis of Fire Extinguishing Efficiency Based on the Average Flame Height

A video software was adopted to process the captured flame image in this study [18]. The video image captured by a high-definition camera was grayed and binarized, and then the flame height data were obtained according to the pixel size, as shown in Figure 3. As the flame height fluctuated greatly, the average flame height data were obtained after smoothing. The change in the average height of the flame at different nitrogen injection flows is depicted in Figure 4. As can be seen from Figure 4, the decreasing trend in the flame height at the initial stage of nitrogen injection varied to a small extent under different test conditions, and the initial suppression of the flame was completed by the nitrogen injection in almost the same period of time. The duration of the flame height fluctuation in conditions 2, 3 and 4 was almost the same, while the fluctuation time in condition 5 was relatively long. It was found that the change in the flow had a limited effect on the flame suppression in the initial stage of the nitrogen injection, but exerted an impact on the maintenance of the low flame height in the later stage of the nitrogen injection. Moreover, the flame was in a constant fluctuation phase without being immediately extinguished, especially when the nitrogen injection flow rate was decreased to a lower value (condition 5).

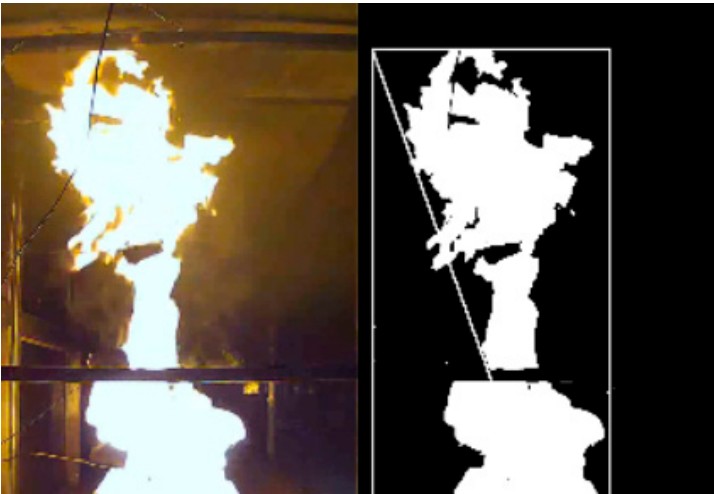

**Figure 3.** Processing method of flame image.

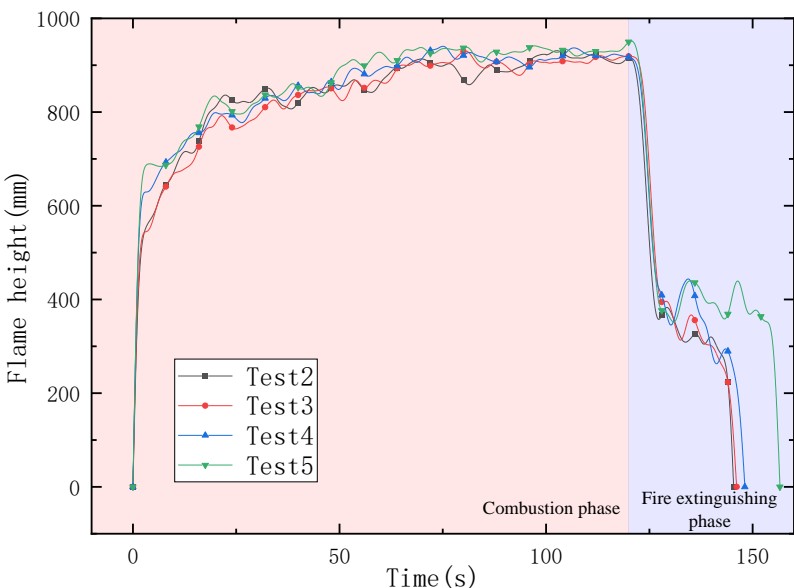

**Figure 4.** Average flame height under different nitrogen injection flow rates.

Based on the test data collected at the beginning and end of the nitrogen injection, the fire suppression factors were calculated under different test conditions, as shown in Table 3. With the continuous increase in the nitrogen injection flow rate, the fire suppression factor showed a continuous upward trend, increasing from the lowest value of 27.88 mm/s to 40.96 mm/s, and the flame suppression effect of the nitrogen injection was also enhanced. In addition, with the increase in the nitrogen injection flow rate, the increase in the range of the fire suppression factor was reduced continuously.

**Table 3.** Fire suppression efficiency under different working conditions.

| Test Number | $Flame_{(t_0)}$ /mm | $Flame_{(t)}$ /mm | Fire Suppression Factor *a*/mm/s |
|:---:|:---:|:---:|:---:|
| 1 | 877.59 | 426.70 | 3.22 |
| 2 | 916.09 | 338.52 | 40.96 |
| 3 | 918.94 | 308.52 | 38.15 |
| 4 | 917.16 | 319.58 | 35.15 |
| 5 | 949.62 | 364.19 | 27.88 |

*4.3. A Comparative Analysis of Fire Extinguishing Efficiency Based on Temperature*

The change in the temperature above the fire source under different nitrogen injection flows is displayed in Figure 5, wherein the moment when the temperature drops to half of the maximum high temperature under different test conditions is denoted by the vertical line. After the beginning of nitrogen injection, the temperature under different test conditions was reduced rapidly, exhibiting a relatively consistent trend. With the increase in the nitrogen injection time, the temperature changes began to differ from one condition to another. For example, the temperature trend in test conditions 2, 3 and 4 was relatively consistent, and the inflection point in the temperature decline appeared after nitrogen injection had stopped. However, no obvious inflection point was observed in the temperature decline curve of test condition 5, and the temperature decreased slowly with time. This was because of the low nitrogen injection flow rate in test condition 5, and the nitrogen accumulation effect generated by the vaporization of liquid nitrogen after injection into the pipe gallery was not obvious. Therefore, the temperature directly above the fire source in condition 5 was barely affected by the interruption of the nitrogen injection.

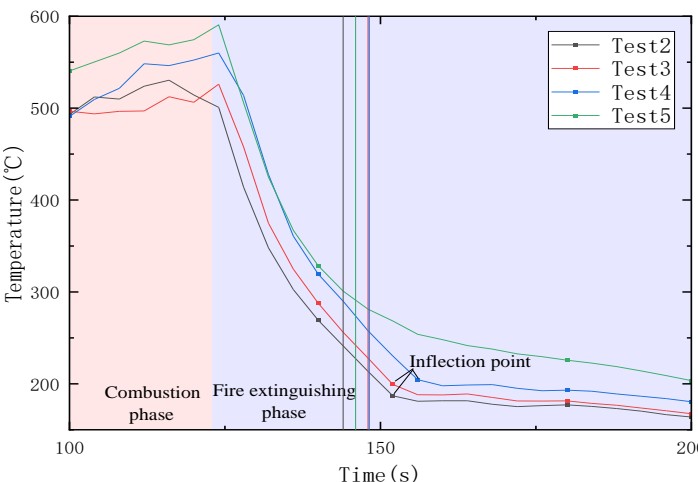

**Figure 5.** Temperature above the fire source under different nitrogen injection flow.

The cooling curves under different test conditions were compared as well, and the cooling law was found to be basically the same in any of the above conditions. The higher the nitrogen injection flow rate, the faster the temperature fell, and the earlier the inflection point appeared in the temperature plot. Meanwhile, the lower the nitrogen injection rate, the slower the temperature dropped. It is worth noting that almost the same downward trend in the initial period of nitrogen injection was shown in the temperatures in conditions 4 and 5, but the difference between them was further widened in the later period of nitrogen injection.

According to the temperature data at the beginning and end of the nitrogen injection, the cooling efficiency of liquid nitrogen under different test conditions was determined, as shown in Table 4. It can be seen that the cooling factor only decreased slightly with the decrease in the nitrogen injection flow rate. In addition, Table 2 shows that under different test conditions, the time needed for the temperature directly above the fire source to drop to half of the maximum temperature was almost the same, and there was a difference between this change rule and the change rule of the fire extinguishing time. Therefore, it was concluded that the flow rate of the nitrogen injection had little influence on the cooling effect and the overall extinguishing efficiency of the nitrogen.

**Table 4.** Cooling efficiency under different test conditions.

| Test Number | $Temperature_{(t_0)}$ /°C | $Temperature_{(t)}$/°C | Cooling Factor $b$/°C/s |
|:-----------:|:-------------------------:|:----------------------:|:-----------------------:|
| 1 | 498.529 | 199.890 | 1.21 |
| 2 | 530.370 | 269.174 | 10.88 |
| 3 | 506.396 | 249.070 | 10.29 |
| 4 | 552.310 | 265.721 | 10.91 |
| 5 | 590.646 | 253.900 | 9.62 |

*4.4. A Comparative Analysis of Fire Extinguishing Effectiveness Based on Oxygen Volume Fraction*

The change in the oxygen volume fraction around the fire source under different nitrogen injection flows is depicted in Figure 6. The time at which the flame was extinguished under different test conditions is denoted by the vertical line. With the change in the nitrogen injection flow, the decreasing trend in the oxygen volume fraction was not significantly different, and the lowest values were relatively close. Under condition 5, due to the longer nitrogen injection time and more liquid nitrogen injected, the oxygen volume fraction was decreased to the largest extent compared to the other values. In the later period of nitrogen injection (before the fire was extinguished), the oxygen volume fraction

decreased slightly faster under the condition of higher nitrogen injection flow, without any obvious differences.

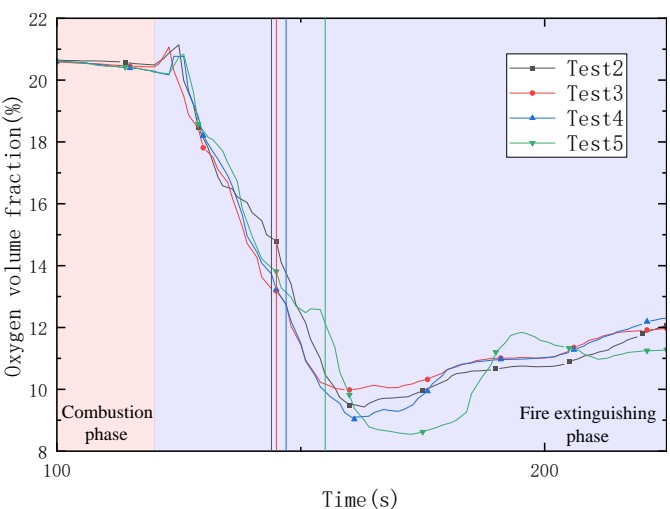

**Figure 6.** Oxygen volume fractions around the fire source under different flow rates.

Furthermore, the nitrogen injection was immediately interrupted after the flame was extinguished, and the oxygen volume fraction under each test condition still maintained a continuous downward trend. This is because the impact of nitrogen injection disappeared after the nitrogen flow was stopped, and the oxygen volume fraction at the bottom was constantly diluted by the nitrogen gas accumulated at the bottom. The difference between the oxygen volume fractions under test conditions 2, 3 and 4 became smaller and started to recover at almost the same time.

The asphyxiation factor of the nitrogen injection under different nitrogen injection flow rates was further calculated according to the oxygen volume fractions at the beginning and end of the nitrogen injection, and the corresponding data are shown in Table 5. It was found that $Oxygen_{(t)}$ decreased with the decrease in the flow rate, which means that more nitrogen accumulated near the fire source with a lower flow rate. There was no obvious law exhibited in regard to the asphyxiation, which is lower at both the higher (condition 2) and lower (condition 5) flow rates. Under condition 5, the nitrogen injection flow was too low, yielding a long fire extinguishing time and a low asphyxiation factor. Under condition 2, due to the overly high nitrogen injection flow rate, the kinetic energy of the nitrogen injection was greater, thereby reducing the nitrogen accumulation near the fire source, which meant that the higher flow was not conducive to the extinguishment. Therefore, it was concluded that either too high or too low nitrogen injection flow can weaken the asphyxiation effect of nitrogen and it should be controlled within an optimal range.

**Table 5.** Asphyxiation efficiency under different test conditions.

| Test Number | $Oxygen_{(t_0)}$/% | $Oxygen_{(t)}$/% | Asphyxiation Factor $c$/%/s |
|:---:|:---:|:---:|:---:|
| 1 | 20.70 | 13.22 | 0.03 |
| 2 | 20.52 | 14.90 | 0.24 |
| 3 | 20.43 | 13.18 | 0.29 |
| 4 | 20.29 | 12.73 | 0.28 |
| 5 | 20.25 | 12.13 | 0.23 |

## 5. The Effect of Injection Pipe Diameter on Liquid Nitrogen Extinguishing Effectiveness

### 5.1. A Comparative Analysis of Fire Extinguishing Times

The fire extinguishing times under different test conditions are shown in Table 6. The extinguishing time in the control group was 246 s, and in condition 6, it was 11 s, which is a reduction of 95.5%. Additionally, the fire extinguishing time presented an increasing trend with a decrease in the pipe diameter.

**Table 6.** Extinguishing time under different test conditions.

| Test Number | Diameter/mm | Extinguishing Time/s |
| --- | --- | --- |
| 1 | - | 246 |
| 6 | 25 | 11 |
| 7 | 20 | 13 |
| 8 | 15 | 14 |
| 2 | 10 | 24 |

### 5.2. A Comparative Analysis of Fire Extinguishing Efficiency Based on the Average Flame Height

The average flame heights at different injection pipe diameters are displayed in Figure 7. The same trend in the flame height was observed within the period of 120 s after ignition. Immediately after starting the nitrogen injection, the flame height was reduced under the impact of liquid nitrogen. The fire was initially controlled by the nitrogen injection for about 5 s and the flame height was reduced from 850 mm to a value between 200 mm and 400 mm. Meanwhile, a rapid decrease in the flame height was recorded when the pipe diameter was larger. Subsequently, the flame height under each test condition entered a short fluctuation phase. It was observed that the larger the pipe diameter, the smaller the duration of the fluctuation phase, indicating that the flame entered the extinguishing phase in a faster manner. For example, test condition 6 had almost no significant fluctuation stage.

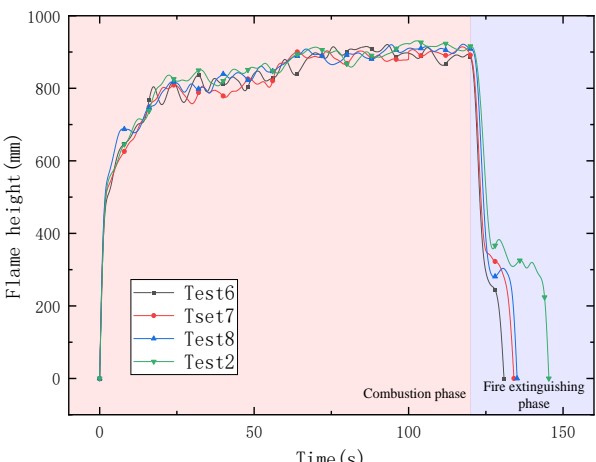

**Figure 7.** Average flame heights at different pipe diameters.

The suppression factors under different test conditions were calculated based on the test data at the beginning and end of the nitrogen injection (see Table 7). The suppression factor showed a continuous downward trend with the decrease in the pipe diameter. In addition, at the large pipe diameter, the flame suppression factor varied greatly according to the pipe diameter, while its change in range was much slower as the pipe diameter was reduced to a certain value.

**Table 7.** Fire suppression efficiency at different pipe diameters.

| Test Number | $Flame_{(t_0)}$ /mm | $Flame_{(t)}$ /mm | Fire Suppression Factor $a$/mm/s |
|:---:|:---:|:---:|:---:|
| 1 | 877.59 | 426.70 | 3.22 |
| 6 | 887.70 | 253.13 | 90.65 |
| 7 | 891.29 | 323.35 | 70.99 |
| 8 | 908.91 | 300.18 | 45.09 |
| 2 | 916.09 | 338.52 | 40.96 |

### 5.3. A Comparative Analysis of Fire Extinguishing Efficiency Based on Temperature

The temperature change in the ceiling above the fire source in the pipe gallery is displayed in Figure 8, and the figure on the right in Figure 8 is a partial enlargement of the left one. The time at which the flame was extinguished under different test conditions is signified by the vertical line. During the early development stage of the flame, there was almost no significant difference between the various test groups, and the temperature was reduced to around 500 °C within 120 s. In the control group, the temperature value at this point continued to fluctuate around 500 °C and then it increased rapidly to around 540 °C at 188 s, followed by a period of accelerated decline until the flame was extinguished (246 s). With the injection of liquid nitrogen, the temperature at this point began to drop rapidly in a linear manner, decreasing to below 200 °C within 20 s.

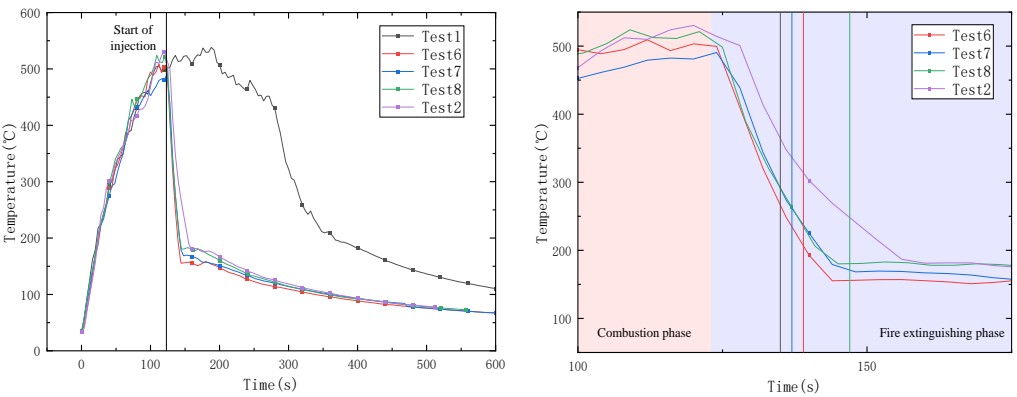

**Figure 8.** Temperatures above the fire source under different conditions.

The cooling factors for the five sets of test conditions are listed in Table 8. On the whole, when the diameter of the nitrogen was decreased, the cooling effect of the nitrogen injection was reduced. Among them, there was an irregular change in test condition 7, and the value of $a$ under test condition 7 showed a sudden decrease from 14.60 °C/s (test condition 6) to 11.89 °C/s. By analyzing $T_{t_0}$, it can be found that $T_{t_0}$ under test condition 7 is lower than that under other test conditions, which may result in the $a$ value under test condition 7 being significantly lower than that under test condition 6 and 8, and this may be caused by experimental errors.

**Table 8.** Nitrogen injection cooling efficiency at different injection pipe diameters.

| Test Number | $Temperature_{(t_0)}$ /°C | $Temperature_{(t)}$/°C | Cooling Factor $b$/°C/s |
|:---:|:---:|:---:|:---:|
| 1 | 498.529 | 199.890 | 1.21 |
| 6 | 503.378 | 342.729 | 14.60 |
| 7 | 481.149 | 326.560 | 11.89 |
| 8 | 518.826 | 320.046 | 14.20 |
| 2 | 530.370 | 269.174 | 10.88 |

### 5.4. A Comparative Analysis of Fire Extinguishing Effectiveness Based on Oxygen Volume Fraction

The change in the oxygen volume fraction around the fire source at different nitrogen injection pipe diameters is depicted in Figure 9, and the figure on the right in Figure 8 is a partial enlargement of the left one. The time at which the flame is extinguished under different test conditions is denoted by the vertical line. In the initial stage of the nitrogen injection, a temporary increase was exhibited in the oxygen volume fraction near the fire source. This is because the newly injected nitrogen had an impact on the flow field in the pipe gallery. Subsequently, under the action of continuous nitrogen injection, the oxygen volume fraction value also continued to decrease around the fire source. Additionally, the oxygen volume fraction near the fire source decreased rapidly with the increase in the nitrogen injection pipe diameter, and the oxygen-isolating asphyxiation effect of nitrogen injection became stronger.

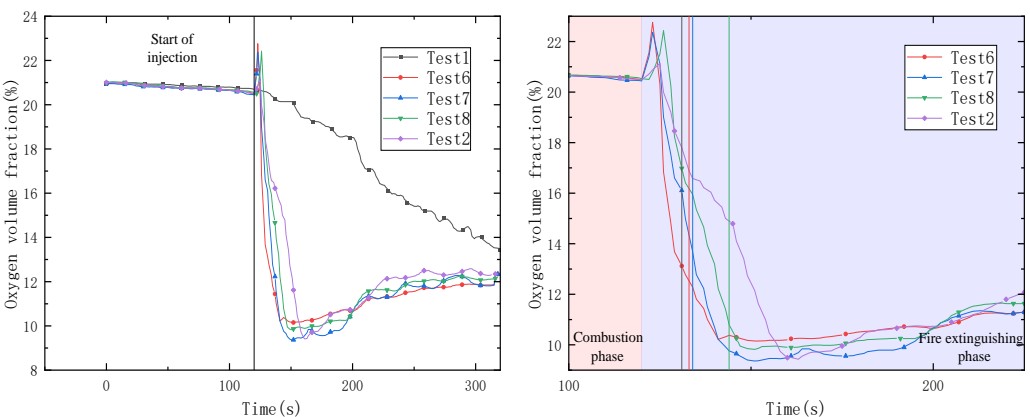

**Figure 9.** Comparison of oxygen volume fractions around the fire source.

According to the oxygen volume fraction data obtained in the test at the beginning and end of the nitrogen injection, the efficiency of the nitrogen asphyxiation at different pipe diameters was determined, as shown in Table 9. As the pipe diameter was reduced, there was a continuous downward trend in the asphyxiation factor, meaning that the asphyxiation effect of the nitrogen injection was positively correlated with the pipe diameter within a certain range. The flow speed of the liquid nitrogen was accelerated because of the decrease in the pipe diameter, which could have aggravated the mixing and disturbance of the liquid nitrogen and the gas around the fire source. As a result, more nitrogen was spread longitudinally along the pipe gallery rather than accumulating at the bottom, so that the asphyxiation effect of nitrogen on the fire source was weakened, thereby reducing the asphyxiation factor to a large extent. When the pipe diameter was reduced from 25 mm to 20 mm and from 20 mm to 15 mm, an isochromatic downward trend was shown in the asphyxiation factor, which indicates that the asphyxiation factor is highly sensitive to the pipe diameter.

**Table 9.** Asphyxiation efficiency of nitrogen injection at different pipe diameters.

| Test Number | $Oxygen_{(t_0)}$/% | $Oxygen_{(t)}$/% | Asphyxiation Factor $c$/%/s |
|:---:|:---:|:---:|:---:|
| 1 | 20.70 | 13.22 | 0.03 |
| 6 | 20.49 | 13.12 | 0.67 |
| 7 | 20.44 | 14 | 0.50 |
| 8 | 20.53 | 15.95 | 0.33 |
| 2 | 20.52 | 14.90 | 0.24 |

**6. Conclusions**

In this study, the fire extinguishing efficiency of liquid nitrogen was investigated by constructing an underground pipe gallery test platform. Special attention was paid to the main fire extinguishing mechanism of liquid nitrogen in underground long and narrow confined spaces, and the main factors affecting fire extinguishing efficiency were examined by varying the nitrogen injection flow and the injection pipe diameter. Based on the findings, the main conclusions are as follows.

- The fire extinguishing performance of liquid nitrogen in long and narrow confined underground spaces was excellent. The shortest fire extinguishing time in the nitrogen injection tests was 11 s, which was 95.5% faster than in the self-extinguishing test.
- The fire suppression efficiency of the nitrogen injection was positively correlated with the nitrogen flow rate, while the cooling efficiency of the nitrogen injection was less affected by the nitrogen flow. The role of cooling in the overall fire extinguishing efficiency of liquid nitrogen was very limited, which proved that asphyxiation was the main fire extinguishing mechanism of liquid nitrogen.
- The nitrogen asphyxiation was reduced when the nitrogen injection rate was too high or too low. Therefore, besides the cost consideration in the engineering application of liquid nitrogen, the most appropriate nitrogen injection flow rate should also be selected to achieve the maximum possible effectiveness of liquid nitrogen fire extinguishment.
- The fire extinguishing efficiency was affected by the change in the injection pipe diameter. In particular, the asphyxiation and fire suppression factors were highly sensitive to the pipe diameter. This also confirmed that asphyxiation was the main fire extinguishing mechanism in liquid nitrogen.

**Author Contributions:** Conceptualization, G.Z.; methodology, G.Z.; software, G.Z.; validation, G.Z.; formal analysis, G.Z. and D.G.; investigation, D.G.; resources, D.G.; data curation, G.Z.; writing—original draft preparation, D.G.; writing—review and editing, B.L. and Z.Z.; visualization, B.L. and Z.Z.; supervision, D.Y.; project administration, G.Z.; funding acquisition, G.Z. All authors have read and agreed to the published version of the manuscript.

**Funding:** This research was funded by Jiangsu Provincial Department of Science and Technology, grant number (BK20221548) and Shenzhen Science and Technology Innovation Commission.

**Institutional Review Board Statement:** Not applicable.

**Informed Consent Statement:** Informed consent was obtained from all subjects involved in the study.

**Data Availability Statement:** No data were used to support this study. However, any query about the research conducted in this paper is highly appreciated and can be asked from the corresponding authors upon request.

**Conflicts of Interest:** The authors declare no conflict of interest.

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
