# Peer review of "An Experimental Investigation of the Influence of Flow and Pipe Diameter on the Fire Extinguishing Efficiency of Nitrogen Injection in a Narrow Confined Underground Space"

_fire, doi:10.3390/fire5060202_

Round 1
Reviewer 1 Report
This paper describes interesting and original work on fire extinguishing with liquid nitrogen in underground enclosed spaces. A thorough literature review was organized in this manuscript to discuss the superiority of liquid nitrogen fire-extinguishing and future research directions of liquid nitrogen fire-extinguishing technology. In addition, the authors experimentally investigated the main fire-extinguishing mechanism of liquid nitrogen in underground enclosed spaces and analyzed the effect of flow/pipe diameter to the fire-extinguishing efficiency of liquid nitrogen from three aspects of fire suppression, cooling and asphyxiation, which attracts my great interest. This study is valuable but needs further improvement as following:
1) Pool fire is not a typical fire scenario of the underground pipe gallery. Please supplement why fuel pool was used as the fire source instead of cables in this study.
2) Was the fire extinguishing started at 120 s under all test conditions? Please supplement the complete process of tests, including when to ignite, when to inject nitrogen, and when to stop the nitrogen injection.
3) There are litter minor editorial issues to fix with spelling typos, e.g., the “he” in the sentence “while he cooling efficiency” in Conclusion should be "the". It is necessary to go over the whole text again.
Author Response
Dear Editor and reviewers:
Thank you again for giving us an opportunity to revise our manuscript. We appreciate very much for the positive and constructive comments and suggestions to improve our manuscript entitled " An experimental investigation of the influence of flow and pipe diameter on the fire extinguishing efficiency of nitrogen injection in a narrow confined underground space ".
We have checked the manuscript and revised it according to the comments. We submit here the revised manuscript. The explanation of what we have changed in response to the reviewers’ comments is given point by point in the following page.
We hope that all these changes fulfil the requirements to make the manuscript acceptable for publication in Fire.
Thank you very much for your attention and consideration.
Sincerely yours,
Zhang Guowei
Comment 1: Pool fire is not a typical fire scenario of the underground pipe gallery. Please supplement why fuel pool was used as the fire source instead of cables in this study.
Reply 1: Thanks for your suggestion. In terms of fire source setting, although cable fire is a common fire scenario of underground pipe gallery, the combustion characteristics of cables are complex. The cable combustion starts for a long time, the development is not obvious, and the transverse and longitudinal propagation speed of combustion is relatively slow, so it is not suitable for use as the test fuel. The main purpose of these tests is to obtain the changes of flame height, temperature, oxygen volume fraction in urban underground pipe gallery under the action of liquid nitrogen. Therefore, oil pool fire with obvious combustion development stage, stable stage and decline stage is selected to simulate the fire scenario in urban underground pipe gallery. The above descriptions have been added to the third paragraph of section 3.
Comment 2: Was the fire extinguishing started at 120 s under all test conditions? Please supplement the complete process of tests, including when to ignite, when to inject nitrogen, and when to stop the nitrogen injection.
Reply 2: Thank you for your suggestion, complete process of tests has been added to the fifth paragraph of section 3.
Comment 3: There are litter minor editorial issues to fix with spelling typos, e.g., the “he” in the sentence “while he cooling efficiency” in Conclusion should be "the". It is necessary to go over the whole text again.
Reply 3: We appreciate your carefulness; the whole test has been fully checked to ensure that such problems will not occur again.

Reviewer 2 Report
See attachment

Author Response
Dear Editor and reviewers:
Thank you again for giving us an opportunity to revise our manuscript. We appreciate very much for the positive and constructive comments and suggestions to improve our manuscript entitled " An experimental investigation of the influence of flow and pipe diameter on the fire extinguishing efficiency of nitrogen injection in a narrow confined underground space ".
We have checked the manuscript and revised it according to the comments. We submit here the revised manuscript. The explanation of what we have changed in response to the reviewers’ comments is given point by point in the following page.
We hope that all these changes fulfil the requirements to make the manuscript acceptable for publication in Fire.
Thank you very much for your attention and consideration.
Sincerely yours,
Zhang Guowei
Comment 1: The setting of the measurement points is not detailed, and the measurement points in Figure 1 are proposed to be arranged in line with the description in the text, and it is proposed to be supplemented.
Reply 1: Thanks for your suggestion. We have updated the Figure 1 and labeled it. Besides, we have added two figures of front view and overhead view to facilitate understanding the test design, especially the sensor arrangement.
Comment 2: How the flame height is measured is suggested to be added.
Reply 2: Thank you for your suggestion, and the measuring method of flame height has been added in the first paragraph of 4.3.
Comment 3: 3. For the description of Figure 5 in 4.3, how can the earlier the inflection point appears in the temperature curve be seen? (Suggested labeling) “Meanwhile…..the temperature drop curve.”(This sentence is not accurately described, suggest re-expression).
Reply 3: We appreciate your very instructive comments. The inflection point has been labelled in Figure 5 and the sentence has been adjusted.
Comment 4: The same working conditions in Figure 8 and Figure 9 are suggested to be represented by the same color curves.
Reply 4: Thank you for your suggestion. The color curves have been modified in Figure 8 and Figure 9 to ensure that readers will not be confused.
Comment 5: 5.3 in the pipe diameter on the cooling effect cannot be said to be no significant effect, from Table 8 can only see no significant law.
Reply 5: Thank you for your preciseness, and the second paragraph of 5.3 has been rewritten.
Comment 6: The horizontal axis time in Figure 9 only goes to about 320 s. It is recommended to synchronize with Figure 8 or shorten the horizontal axis.
Reply 6: Thank you for your carefulness, and the horizontal axis in Figure 9 has been shortened.
Comment 7: It is recommended that the relationship between fire extinguishing factor, cooling factor, and asphyxiation factor be properly summarized at the end of different nitrogen injection volumes and different jet pipe diameter sections (end of sections 4 and 5).
Reply 7: Thank you for your advice. The relationship between fire extinguishing factor, cooling factor, and asphyxiation factor is quite important and worthy analyzing for the determination of the main fire extinguishing mechanism of liquid nitrogen in the underground pipe gallery. While it is a hard work and need more test data to carry out the research. Actually, three factors are used to evaluate the effect of nitrogen injection flow rate and pipe diameter to the fire extinguishing efficiency in the underground pipe gallery. We think your suggestion very instructive, and we will study this relationship in the next stage of work.

Reviewer 3 Report
An experimental investigation of influence of flow and pipe diameter on fire extinguishing efficiency of nitrogen injection in a narrow confined underground space
The authors have experimentally investigated the impact of nitrogen injection on fire extinguishment in a simulated narrow confined underground space. The authors performed experiments in a setup and have presented results of temperature, flame height and derived efficiency. The paper, however, lacks in scientific writing, incomplete and hard to understand experimental setup, conclusion from the obtained data. The results from the research could be important, but only when this manuscript is updated substantially. Below are my comments, which should be addressed to improve the manuscript to an acceptable level.
- The paper has no page numbers or line numbers. Although the manuscript will be stylized according to the journal, it is hard to read such manuscript.
- Section 1, rephrase “in this situation, the fire extinguishing efficiency is suppressed”. Efficiency will either reduce or improve but will not suppress.
- “Nitrogen is a non-toxic and harmless green gas”. At least, remove the word “green” from this sentence. In this context it is not easy to understand that the authors want to highlight the zero global warming potential of nitrogen (I am assuming that’s what this means).
- “Lan et al. reported that Accelerated smoke migration..”. This sentence is confusing. Was the smoke forced out of the confined space?
- Examples of writing that does not meet the criteria of a scientific literature and highlight the lack of repetitive quantitative assessment of the data.
o “but would expand the spread range of smoke”
o “extinguishing ability of liquid nitrogen was weakened to a certain extent at the low flow rate”
- Experimental setup and measurements lack sufficient information for validating and reproducing the results.
o Where was the oxygen concentration measured? Why wherever it was measured? How close was it to the nitrogen injection pipe?
o Various factors used to assess suppression assume linearity, which is fine for the a first order estimate. However, none of these factors consider the mass of the nitrogen as a parameter. It seems that efficiency of an extinguishing agent in terms of per mass used is crucial. Although not all the extinguishing agent will directly interact to suppress the fire, this information could assist design and management of equipment that could employ nitrogen extinguishing agents.
o How do the various factors considered here compare with efficiency of other extinguishing agents?
o How was the time of fire extinguishment determined? It is unclear how this was obtained and seems arbitrary.
- Test design
o First, fig. 1 has absolutely no labels on it. It is unclear figure with unnecessary details of color or rolling casters. What is essential in this figure is to show the exact layout of the pipe gallery, fuel pan, instruments. A 2-D figure, maybe from different orientations can be helpful. I could not understand, even after reading the manuscript, location of important instruments mentioned in the manuscript.
o The beginning of the first two paragraphs of Test Design section seems repetitive.
o Why was n-heptane used in these experiments? It probably creates a sootier flame than other fuels.
o How/why did the fire self-extinguish in the “control group” test condition?
o I recommend to use either “suffocation” or “asphyxiation” in the entire manuscript to be consistent.
o How was the nitrogen injection pattern? What was the intention of this pattern in the experiments?
o How representative is the pool fire for an underground fire scenario?
o You vaguely mention that you measured “other parameters” as if discussing this was not important.
o Use a technical term than a “heyday stage”
o Meanwhile, “flue gas…” what is the flue gas here? Nothing about this is discussed in the paper.
o How many times was each experiment conducted? The reputability of the experiments is not discussed at all.
- You do not mention the actual flowrates of nitrogen in the test condition, but a fraction of maximum flow rate. I think supporting the data with actual flow rate information and associated pressure is important to assess extinguishing efficiency.
- Was the maximum flow rate for smaller diameter essentially equivalent to flow rate with larger diameter and a fractional flow? If that’s the case, how close were the extinguishing times for these two scenarios? As mentioned earlier, mass flow rate should be considered to determine suppression efficiency.
- The tabulated results have table headers which are hard to understand, and the variables have no reference to what they mean.
- As the author mention in the conclusion that too high or too low nitrogen flow would not cause extinguishment, the range of flow rates and pressure for which extinguishment occurs could be obtained from careful experimental data analysis.
Overall, this paper requires additional work, in order to be considered for publication.
Author Response
Dear Editor and reviewers:
Thank you again for giving us an opportunity to revise our manuscript. We appreciate very much for the positive and constructive comments and suggestions to improve our manuscript entitled " An experimental investigation of the influence of flow and pipe diameter on the fire extinguishing efficiency of nitrogen injection in a narrow confined underground space ".
We have checked the manuscript and revised it according to the comments. We submit here the revised manuscript. The explanation of what we have changed in response to the reviewers’ comments is given point by point in the following page.
We hope that all these changes fulfil the requirements to make the manuscript acceptable for publication in Fire.
Thank you very much for your attention and consideration.
Sincerely yours,
Zhang Guowei
Comment 1: The paper has no page numbers or line numbers. Although the manuscript will be stylized according to the journal, it is hard to read such manuscript.
Reply 1: Sorry for the bad reading experience. Page numbers have been added to the text.
Comment 2: Section 1, rephrase “in this situation, the fire extinguishing efficiency is suppressed”. Efficiency will either reduce or improve but will not suppress.
Reply 2: Sorry about my English writing, and “suppressed” in this sentence has been replaced by “reduced”.
Comment 3: “Nitrogen is a non-toxic and harmless green gas”. At least, remove the word “green” from this sentence. In this context it is not easy to understand that the authors want to highlight the zero global warming potential of nitrogen (I am assuming that’s what this means).
Reply 3: As you said, “the zero global warming potential of nitrogen” is exactly what we want to highlight. It is maybe a Chinese English expression and we did not realize that it is hard for understand. The word “green” has been removed from this sentence. Thanks for your advice.
Comment 4: “Lan et al. reported that Accelerated smoke migration..”. This sentence is confusing. Was the smoke forced out of the confined space?
Reply 4: Sorry for the misunderstanding, and what we want to express is that ventilation will have both positive and negative effects on fire extinguishing by nitrogen injection. The original sentence has been rewritten as “Lan et al. reported that ventilation in the confined space could enhance the fire extinguishing efficiency of liquid nitrogen to a certain extent, but would also make the smoke spread further within the same time”. We have also revised it in the text, and thanks for your guidance.
Comment 5: Examples of writing that does not meet the criteria of a scientific literature and highlight the lack of repetitive quantitative assessment of the data, e.g., “but would expand the spread range of smoke” and “extinguishing ability of liquid nitrogen was weakened to a certain extent at the low flow rate”.
Reply 5: Thank you for your professional guidance. We have revised these two examples you mentioned, and we have also checked the whole text again to reduce such unscientific and unprofessional descriptions.
Comment 6: Experimental setup and measurements lack sufficient information for validating and reproducing the results. Where was the oxygen concentration measured? Why wherever it was measured? How close was it to the nitrogen injection pipe?
Reply 6: Thanks for your guidance. The whole Test Design has been updated and supplemented. The oxygen volume fraction sensor was placed at the same height as the fuel pan, and 0.1 m away from the fuel pan. The reason why wherever it was placed is that we wanted to evaluate the asphyxiation effect of nitrogen injection to the fire source and also had to avoid the damage and accuracy decline due to the high temperature. Besides, in order to avoid the direct impact of nitrogen injection to the sensor, it was placed at another side of the fire source. The horizonal distance between the sensor and the nitrogen injection port was nearly 0.75 m, and the vertical distance was about 1.05 m. The figure and description of sensor arrangement have been supplemented in Test Design.
Comment 7: Various factors used to assess suppression assume linearity, which is fine for the first order estimate. However, none of these factors consider the mass of the nitrogen as a parameter. It seems that efficiency of an extinguishing agent in terms of per mass used is crucial. Although not all the extinguishing agent will directly interact to suppress the fire, this information could assist design and management of equipment that could employ nitrogen extinguishing agents.
Reply 7: I totally agree with your point of view, which provides a powerful guidance for our follow-up research. The fire extinguishing efficiency of liquid nitrogen in terms of per mass is indeed an important evaluation parameter. Actually, in the previous tests, we tried to calculate the consumption data of liquid nitrogen by the real-time flow rate and total nitrogen injection time, while we found that the data were hardly usable due to the characteristics of two-phase flow of nitrogen in the pipeline. Placing a high-precision balance underground the liquid nitrogen tank seems more feasible. But after the test, there will be a lot of nitrogen left in the pipeline, and we cannot obtain the amount of liquid nitrogen that was really injected into the pipe gallery. We will continue to find appropriate means to achieve this in the future research. We’re looking forward to your professional guidance in terms of this issue. Many thanks.
Comment 8: How do the various factors considered here compare with efficiency of other extinguishing agents?
Reply 8: We preliminarily proved the fire extinguishing efficiency of liquid nitrogen in the previous experimental study. The purpose of this study is to explore the influence of different parameters on the fire extinguishing effect of liquid nitrogen in the underground pipe gallery. Of course, what you said is quite correct, which is essential to prove the superiority of the liquid nitrogen fire extinguishing efficiency in the underground pipe gallery. We will carry out the comparison tests of different fire extinguishing agents in the subsequent research.
P.S. Our previous research:
1.Experimental investigation on extinguishing characteristics of liquid nitrogen in underground long and narrow space - ScienceDirect
2.Influence of injection method on the fire extinguishing efficiency of liquid nitrogen in urban underground utility tunnel - ScienceDirect
3.Applicability of liquid nitrogen fire extinguishing in urban underground utility tunnel -ScienceDirect
Comment 9: How was the time of fire extinguishment determined? It is unclear how this was obtained and seems arbitrary.
Reply 9: We added complete process of tests to the fifth paragraph of section 3. In our test design, we can observe flames through the high-definition camera and the thermal imaging camera inside the pipe gallery. N-heptane is a relatively clean fuel, so it doesn't produce too much black smoke. And Thermal imaging camera can precisely capture when flame goes out. Thus, we were able to capture exactly when the flame went out during the test, and recorded the time. We define the time from the beginning of nitrogen injection to the time of flame extinction as the time of fire extinguishment. We have also supplemented it in the first paragraph of 4.1, and thanks for your guidance.
Comment 10: First, fig. 1 has absolutely no labels on it. It is unclear figure with unnecessary details of color or rolling casters. What is essential in this figure is to show the exact layout of the pipe gallery, fuel pan, instruments. A 2-D figure, maybe from different orientations can be helpful. I could not understand, even after reading the manuscript, location of important instruments mentioned in the manuscript.
Reply 10: Thanks for your professional guidance. The previous figures are not clear enough indeed, and too difficult to understand. Following your suggestions, we have updated the Figure 1 and labeled it. Besides, we have added two figures of front view and overhead view to facilitate understanding the test design.
Comment 11: The beginning of the first two paragraphs of Test Design section seems repetitive.
Reply 11: We have rewritten the text of 3.1, deleted the repetition part, and added the description to make the test design more detailed.
Comment 12: Why was n-heptane used in these experiments? It probably creates a sootier flame than other fuels.
Reply 12: Although n-heptane is not a completely clean fuel, we found that the smoke generated by n-heptane is acceptable and will not affect the flame observation through early testing. Compared with n-heptane, methanol fuel is cleaner, but its combustion heat is lower and its flame color is lighter, which will affect the flame observation. Therefore, considering comprehensively, we chose n-heptane to carry out the research. The description of n-heptane in the article has also been supplemented.
Comment 13: How/why did the fire self-extinguish in the “control group” test condition?
Reply 13: In this study, the research focus is to compare the effects of different nitrogen injection flow rates and nitrogen injection pipe diameters on nitrogen injection fire extinguishing efficiency, which means that it may not be necessary to set up a control group. We added the self-extinguishing test to briefly show the fire extinguishing effect of liquid nitrogen and did not compare the control group with other groups too much. As is mentioned in Reply 8, We will carry out the comparison tests of different fire extinguishing agents in the subsequent research.
Comment 14: I recommend to use either “suffocation” or “asphyxiation” in the entire manuscript to be consistent.
Reply 14: Thanks for your advice, and all “suffocation” has been replaced with “asphyxiation” in the text.
Comment 15: How was the nitrogen injection pattern? What was the intention of this pattern in the experiments?
Reply 15: We have added the description of nitrogen injection pattern in the fifth paragraph of section 3.1. In the previous study, we compared the effect of different nitrogen injection directions and different nitrogen injection distances to the fire extinguishing efficiency of liquid nitrogen in the underground pipe gallery, and found that vertical downward nitrogen injection was better than horizontal nitrogen injection, and the closer the nitrogen injection distance, the better fire extinguishing efficiency. Thus, in this study, wo chose vertical downward nitrogen injection and 0.3 m nitrogen injection distance as the nitrogen injection pattern, and carried out tests by changing the flow rate and pipe diameter.
P.S. Our previous research:
1.Experimental investigation on extinguishing characteristics of liquid nitrogen in underground long and narrow space - ScienceDirect
2.Influence of injection method on the fire extinguishing efficiency of liquid nitrogen in urban underground utility tunnel - ScienceDirect
3.Applicability of liquid nitrogen fire extinguishing in urban underground utility tunnel -ScienceDirect
Comment 16: How representative is the pool fire for an underground fire scenario?
Reply 16: In terms of fire setting in the test, although cable fire is a common fire scenario of underground pipe gallery, the combustion characteristics of cables are complex. The cable combustion starts for a long time, the development is not obvious, and the transverse and longitudinal propagation speed of combustion is relatively slow, so it is not suitable for use as the test fuel. The main purpose of this research is to obtain the changes of flame height, temperature, oxygen volume fraction in urban underground pipe gallery under the action of liquid nitrogen. Therefore, pool fire with obvious combustion development stage, stable stage and decline stage is selected to simulate the fire scenario in urban underground pipe gallery. Of course, for the engineering application of liquid nitrogen in underground pipe gallery fire, it is necessary to test the fire extinguishing effect of liquid nitrogen on cable fire, which will be carried out later in our research plan. The above descriptions have been added to the third paragraph of section 3.
Comment 17: You vaguely mention that you measured “other parameters” as if discussing this was not important.
Reply 17: Actually, in addition to the flame height, temperature and oxygen volume fraction, we also captured some other test data. For example, a mass sensor was placed under the oil pool, which can obtain the mass data of the fuel during the test, and then we can calculate the heat release rate of combustion. But these data are not used in this article. In order not to cause misunderstanding, we deleted “other parameters”. Thanks for your attention.
Comment 18: Use a technical term than a “heyday stage”.
Reply 18: Thanks for your professional guidance, the “heyday stage” has been replaced with “fully developed fire stage”.
Comment 19: Meanwhile, “flue gas…” what is the flue gas here? Nothing about this is discussed in the paper.
Reply 19: Sorry for the misunderstanding. I mistakenly believe that “flow gas” equals “smoke”, and actually I want to express “smoke”. The “flue gas…” in the text have been replaced with “smoke”.
Comment 20: How many times was each experiment conducted? The reputability of the experiments is not discussed at all.
Reply 20: Sorry for the incomplete description about the experiment. Three repeated tests were conducted for the test condition 2 and the time of fire extinguishment were 24 s, 23 s and 29 s. A good repeatability of the test result was obtained. For other tests, one test was conducted. The reputability of the experiments has also been supplemented in the section 3.2.
Comment 21: You do not mention the actual flowrates of nitrogen in the test condition, but a fraction of maximum flow rate. I think supporting the data with actual flow rate information and associated pressure is important to assess extinguishing efficiency.
Reply 21: Thanks for your advice. The use of a fraction rather the actual flowrate is to consider that it is more intuitive. We did not measure the pressure value corresponding to the flowrate. The pressure gauge in the liquid nitrogen tank only displays the pressure inside the tank. We have supplemented the maximum flow rate in the first paragraph of section 3.2. And in the subsequent study, we will take your advice to measure the pressure.
Comment 22: Was the maximum flow rate for smaller diameter essentially equivalent to flow rate with larger diameter and a fractional flow? If that’s the case, how close were the extinguishing times for these two scenarios? As mentioned earlier, mass flow rate should be considered to determine suppression efficiency.
Reply 22: Thank you for your question. Actually, the maximum flow rate for smaller diameter does not equivalent to flow rate with larger diameter and a fractional flow. This misunderstanding was caused by the unclear description in the text. The flow rate was obtained from the Coriolis Mass Flowmeter and was not in the nitrogen injection port. We have updated Figure 1 to make the location of the flowmeter clearer. And as is mentioned in the Reply 7, we will try to consider fire extinguishing efficiency of liquid nitrogen in terms of per mass in the future research.
Comment 23: The tabulated results have table headers which are hard to understand, and the variables have no reference to what they mean.
Reply 23: Sorry for the bad review experience, we have updated all tables to make sure they are easy for reading. And we have supplemented the description of variables in the third paragraph of section 2.
Comment 24: As the author mention in the conclusion that too high or too low nitrogen flow would not cause extinguishment, the range of flow rates and pressure for which extinguishment occurs could be obtained from careful experimental data analysis.
Reply 24: Your comment is quite right; too high or too low nitrogen flow is not conducive to fire extinguishment and it is necessary for more analysis. The optimal flow rate and pressure can be obtained by carrying out more tests, which we will conduct in the future research.

Reviewer 4 Report
While the age of digital publication has made virtually all honest, accurately measured results worth reporting, the results were entirely intuitively, and only weakly applicable to an extrapolation beyond the actual experimental apparatus into a real-world engineering use. The analysis was trivial, further exacerbating any potential real-world extrapolation difficulties.
Author Response
Dear Editor and reviewers:
Thank you again for giving us an opportunity to revise our manuscript. We appreciate very much for the positive and constructive comments and suggestions to improve our manuscript entitled " An experimental investigation of the influence of flow and pipe diameter on the fire extinguishing efficiency of nitrogen injection in a narrow confined underground space ".
Fire in the long and narrow underground space is a difficult problem to deal with, and some new fire extinguishing technologies are needed. Applying liquid nitrogen to extinguish fire in underground confined space is our team's main research work in recent years. The team's previous research also confirmed that liquid nitrogen fire extinguishing is efficient, and very suitable for the underground pipe gallery, cable trench and other scenes. In order to promote the engineering application of liquid nitrogen, we put forward an evaluation index of liquid nitrogen fire extinguishing efficiency, and explored the parameters affecting liquid nitrogen fire extinguishing efficiency through three factors.
Your comments and criticisms are very important, and let us realize that there is still much room for improvement in our article. We have checked the manuscript and revised it according to the comments substantially. What we have changed has been marked red in the text. At the same time, we will continue to complete our research design and analysis in the future work to make our research more scientific and rigorous.
We hope that all these changes fulfil the requirements to make the manuscript acceptable for publication in Fire.
Thank you very much for your attention and consideration.
Sincerely yours,
Zhang Guowei

Round 2
Reviewer 2 Report
Fig. 8 and Fig. 9 in the right figure is a partial enlargement of the left figure suggested for illustration to increase readability.
Author Response
Dear Editor and reviewers:
We appreciate your recognition and suggestions for our manuscript. Your comments are essential to the improvement of our article. We improved the article according to the comments of the reviewers, hoping to meet the requirements of publishing in Fire. The explanation of what we have changed in response to the reviewers’ comments is given point by point in the following page. All the contents revised in the second round are marked in blue in the text.
Thank you very much for your attention and consideration.
Sincerely yours,
Zhang Guowei
Comment 1: Fig. 8 and Fig. 9 in the right figure is a partial enlargement of the left figure suggested for illustration to increase readability.
Reply 1: Thanks for your suggestion. The description of Fig. 8 and Fig. 9 is necessary to increase readability. Corresponding description has been added in the first paragraphs of section 5.3 and 5.4.

Reviewer 3 Report
Thanks for sending the revised document. The changes in the test diagram is understandable and valuable now. This document is detailed than the previous manuscript but can still be improved further, especially the english.
- Some of the edited sentences are now active voices, but the remainder of the document is passive. for example section 3, paragraph 2. There is also one sentence "Meanwhile, the smoke has filled half of the pipe gallery. ". Such sentences must be improved. The entire manuscript must be changed to maintain the correct voice, preferably passive. Correcting these two sentences is not sufficient and reading the entire manuscript, sentence by sentence, is necessary and important next step.
- The answer to as to why and how the fire self-extinguished in the control group is not yet clear.
Thank you
Author Response
Dear Editor and reviewers:
We appreciate your recognition and suggestions for our manuscript. Your comments are essential to the improvement of our article. We improved the article according to the comments of the reviewers, hoping to meet the requirements of publishing in Fire. The explanation of what we have changed in response to the reviewers’ comments is given point by point in the following page. All the contents revised in the second round are marked in blue in the text.
Thank you very much for your attention and consideration.
Sincerely yours,
Zhang Guowei
Comment 1: Some of the edited sentences are now active voices, but the remainder of the document is passive. for example section 3, paragraph 2. There is also one sentence "Meanwhile, the smoke has filled half of the pipe gallery. ". Such sentences must be improved. The entire manuscript must be changed to maintain the correct voice, preferably passive. Correcting these two sentences is not sufficient and reading the entire manuscript, sentence by sentence, is necessary and important next step.
Reply 1: Thank for your professional guidance. The manuscript has been revised and the active voices was modified into the passive voices as much as possible. The corresponding modifications in manuscript are marked in blue. We very much hope there are no omissions or improper modifications.
Comment 2: The answer to as to why and how the fire self-extinguished in the control group is not yet clear.
Reply 2: Thank your question. For the enclosed space, the oxygen content is the key to the fire occurrence and development. Many enclosed space fires, when human control is not possible, usually extinguish themselves after the oxygen is exhausted. And for fire in enclosed space, asphyxiation is a very effective fire extinguishing method. At the same time, liquid nitrogen is a fire extinguishing agent with excellent asphyxiation ability. In the fire scenario of underground enclosed space, in most cases, fire fighters cannot directly aim at the fire source to extinguish the fire. Instead, using the mode of massive injection of liquid nitrogen to dilute the internal oxygen concentration is more feasible. Thus, considering the fire characteristics of the enclosed space and the fire extinguishing characteristics of liquid nitrogen, we selected the self-extinguishing group as a comparison to show the fire extinguishing efficiency of liquid nitrogen.
